# Dynamic Monitoring of Intracellular Tacrolimus and Mycophenolic Acid Therapy in Renal Transplant Recipients Using Magnetic Bead Extraction Combined with LC-MS/MS

**DOI:** 10.3390/pharmaceutics15092318

**Published:** 2023-09-14

**Authors:** Huan Xu, Yingying Liu, Yinan Zhang, Xinhua Dai, Xueqiao Wang, Haojun Chen, Lin Yan, Xingxin Gong, Jiaxi Yue, Zhengli Wan, Jiwen Fan, Yangjuan Bai, Yao Luo, Yi Li

**Affiliations:** 1Department of Laboratory Medicine, West China Hospital, Sichuan University, Chengdu 610041, China; xhhuan211@163.com (H.X.); daixinhua@scu.edu.cn (X.D.); wangsled@163.com (X.W.); yanlinscuedu@outlook.com (L.Y.); gongxingxin2021@163.com (X.G.); jiaxi990720@163.com (J.Y.); 15984500518@163.com (Z.W.); fjw707126427@outlook.com (J.F.); whitewcums@163.com (Y.B.); 2School of Chemistry and Chemical Engineering, Frontiers Science Center for Transformative Molecules and National Center for Translational Medicine (Shanghai), Shanghai Jiao Tong University, Shanghai 200240, China; yangyang21@sjtu.edu.cn; 3School of Chemical Science and Engineering, Tongji University, Shanghai 200092, China; yinan_zhang@tongji.edu.cn; 4Department of Laboratory Medicine, West China Fourth Hospital, Sichuan University, Chengdu 610041, China; chenhaojun1059@163.com

**Keywords:** renal transplant, tacrolimus, mycophenolic acid, peripheral blood mononuclear cell, LC-MS/MS

## Abstract

Background: Tacrolimus (TAC) and mycophenolic acid (MPA) are commonly used immunosuppressive therapies after renal transplant. Our objective was to quantify TAC and MPA concentrations in peripheral blood mononuclear cells (PBMCs) using liquid chromatography tandem mass spectrometry (LC-MS/MS) and to evaluate and validate the performance of the methodology. A prospective follow-up cohort study was conducted to determine whether intracellular concentrations were associated with adverse outcomes in renal transplants. Methods: PBMCs were prepared using the Ficoll separation technique and purified with erythrocyte lysis. The cells were counted using Sysmex XN-3100 and then packaged and frozen according to a 50 µL volume containing 1.0 × 10^6^ cells. TAC and MPA were extracted using MagnaBeads and quantified using an LC-MS/MS platform. The chromatography was run on a reversed-phase Waters Acquity UPLC BEH C18 column (1.7 µm, 50 mm × 2.1 mm) for gradient elution separation with a total run time of 4.5 min and a flow rate of 0.3 mL/min. Mobile phases A and B were water and methanol, respectively, each containing 2 mM ammonium acetate and 0.1% formic acid. Renal transplant recipients receiving TAC and MPA in combination were selected for clinical validation and divided into two groups: a stable group and an adverse outcome group. The concentrations were dynamically monitored at 5, 7, 14, and 21 days (D5, D7, D14, and D21) and 1, 2, 3, and 6 months (M1, M2, M3, and M6) after operation. Results: Method performance validation was performed according to Food and Drug Administration guidelines, showing high specificity and sensitivity. The TAC and MPA calibration curves were linear (r^2^ = 0.9988 and r^2^ = 0.9990, respectively). Both intra-day and inter-day imprecision and inaccuracy were less than 15%. Matrix effects and recoveries were satisfactory. The TAC and MPA concentrations in 304 “real” PBMC samples from 47 renal transplant recipients were within the calibration curve range (0.12 to 16.40 ng/mL and 0.20 to 4.72 ng/mL, respectively). There was a weak correlation between PBMC-C_0TAC_ and WB-C_0TAC_ (*p* < 0.05), but no correlation was found for MPA. The level of immunosuppressive intra-patient variation (IPV) was higher in PBMC at 77.47% (55.06, 97.76%) than in WB at 34.61% (21.90, 49.85%). During the dynamic change in C_0TAC_, PBMC-C_0TAC_ was in a fluctuating state, and no stable period was found. PBMC-C_0TAC_ did not show a significant difference between the stable and adverse outcome group, but the level of the adverse outcome group was generally higher than that of the stable group. Conclusions: Compared with conventional therapeutic drug monitoring, the proposed rapid and sensitive method can provide more clinically reliable information on drug concentration at an active site, which has the potential to be applied to the clinical monitoring of intracellular immunosuppressive concentration in organ transplantation. However, the application of PBMC-C_0TAC_ in adverse outcomes of renal transplant should be studied further.

## 1. Introduction

Renal transplant recipients undergo lifelong maintenance of immunosuppressive therapy, and the balance between long-term management of immunosuppressive toxicity and the risk of chronic antibody-mediated rejection remains complex. The vast majority of patients receive the calcineurin inhibitor tacrolimus (TAC) in combination with mycophenolic acid (MPA) and corticosteroids as the main drug therapy [1]. MPA exerts an immunosuppressive effect by blocking the synthesis of guanosine and deoxyguanosine nucleotides in T and B lymphocytes, inhibiting cell proliferation and cell-mediated immune responses and antibody formation [2,3]. The immunosuppressive effect of TAC is to inhibit the expression of cytokines necessary for immune activation, such as interleukin-2 (IL-2), by inhibiting the activation of calcineurin in lymphocytes [4,5]. To achieve an immunosuppressive effect, TAC and MPA mainly target lymphocytes.

Although the use of immunosuppressants is effective in preventing acute rejection, clinical management is still difficult due to the narrow therapeutic window and the large variability in pharmacokinetic (PK) and pharmacodynamic (PD) characteristics between individuals [6]. Therefore, therapeutic drug monitoring (TDM) is routinely performed in the clinic to maintain target ranges and avoid over- or under-exposure. Trough concentrations (C_0_) have been widely used as a guide for the individualization of immunosuppressive doses. However, it is worth noting that rejection and toxicity still occurred when whole blood trough concentrations (WB-C_0_) were maintained within the normal dose range [7,8]. In the blood, TAC is primarily associated with erythrocytes, followed by diluted plasma proteins and lymphocytes [9]. Therefore, routine monitoring of WB-C_0_ cannot reflect the amount of drug concentration in target cells that actually achieves drug efficacy.

Peripheral blood mononuclear cells (PBMCs) are lymphocyte- and monocyte-rich cell populations. Capron, A. et al. found that the TAC concentration in liver tissue after liver transplantation was significantly correlated with the severity of organ rejection compared with C_0_ in whole blood (WB-C_0_) [10]. Capron, A. et al. further conducted a prospective study to evaluate PBMC TAC levels as a predictive value for early efficacy after liver transplantation, which appeared to be easier than tissue drug measurements requiring invasive biopsy [11]. Their findings suggested that PBMC TAC C_0_ (PBMC-C_0TAC_) may be a reliable marker of early immunosuppression after liver transplantation and that PBMC may represent an additional tool for more precise individualized early immunosuppression regimens after liver transplantation. Similarly, the concentration of MPA in PBMC was associated with the incidence of rejection in renal transplant recipients, and its clinical application may be most practical as a single test in the early post-transplant period [12]. A population PK model in renal transplant recipients demonstrated the intracellular accumulation of MPA, but the correlation between PBMC MPA C_0_ (PBMC-C_0MPA_) and optimal administration regimen needs further study [13]. PBMCs have been used as effective matrices for quantifying intracellular immunosuppressant concentrations [12,14,15,16].

At present, the quantitative application of intracellular immunosuppressant concentration in organ transplantation has attracted more and more attention, which is expected to achieve individualized precision treatment. TAC, MPA, and hormone combined therapy is the most commonly used immunosuppressive regimen for anti-allograft rejection [6,17] and is the most common immunosuppressive regimen in the kidney transplant center of our hospital. Intracellular immunosuppressive levels require more sensitive analysis than conventional immunological methods. LC-MS/MS is used to quantify the concentration of a variety of immunosuppressants, including TAC, MPA, cyclosporine A (CsA), sirolimus, and everolimus [16,18,19,20], but clinical applications are limited due to the complex and time-consuming pre-processing of clinical samples. However, the magnetic bead method for extracting intracellular drug concentration is more popular because of its convenience, speed, and good operation [21].

Our study aimed to establish an LC-MS/MS experimental research scheme in our laboratory, including the use of magnetic beads to extract drugs from PBMC samples and the separation and analysis of drugs using chromatography and mass spectrometry. We then designed a prospective follow-up cohort study including a group of kidney transplant recipients to explore whether the risk of transplant rejection and opportunistic infection after renal transplant is more associated with intracellular immunosuppressive concentrations.

## 2. Materials and Methods

### 2.1. Chemicals, Reagents, and PBMCs from Healthy Volunteers

TAC, MPA, and mycophenolic acid-d3 (MPA-d3) which is a deuterated product of mycophenolic acid, were purchased from Toronto Research Chemicals (Toronto, ON, Canada), and [^13^C, ^2^H_4_]-tacrolimus was purchased from Alsachim (Strasbourg, France). Methanol, acetonitrile, and formic acid were obtained from Fisher Chemical (Thermo Scientific, Waltham, MA, USA) at ultra-high performance liquid chromatography/mass spectrometry grades. Ammonium acetate was obtained from Sigma-Aldrich (St. Louis, MO, USA). Ultra-pure water was supplied by the Milli-Q integral water purification system (Merck Millipore, Germany). MagSiMUS-TDM^prep^ Type I Particle Mix, Organic Precipitation Reagent VI (OPR VI), and Lysis Buffer for whole blood kits were purchased from MagnaMedics Diagnostics B.V. (Geleen, The Netherlands). Ficoll-Paque Plus solution, Dulbecco’s phosphate-buffered saline (D-PBS) without calcium and magnesium ions, and red blood cell lysis buffer were obtained from Solarbio (Beijing, China). PBMCs for the establishment and validation of analytical methods were obtained from healthy volunteers in agreement with the local ethics committee.

### 2.2. Isolation of PBMCs

The protocol for isolation of blank PBMCs used for the preparation of standards, quality controls (QCs), and patient PBMCs was the same. Blank PBMCs were isolated from approximately 100 mL of fresh EDTA-K_2_-anticoagulated whole blood (WB) from 40 healthy volunteers. The blood was diluted by adding D-PBS equal to the volume of WB. Then, the mixture was slowly tiled over Ficoll-Paque Plus solution of equal volume while the liquid in the 15 mL polypropylene tube remained stratified without sloshing, which was then centrifuged at 500× *g* for 20 min in the non-brake mode (5810, Eppendorf Company, Hamburg, Germany). The cells in the PBMC layer were transferred to D-PBS for washing and then centrifuged at 300× *g* for 10 min. To reduce the presence of red blood cells, the PBMC suspensions were dissolved with red blood cell lysate for 10 min and centrifuged at 300× *g* for 10 min. PBMC suspensions were again treated with D-PBS and erythrocyte lysate, and cells were finally re-suspended with D-PBS. Cell suspensions were removed for cell counting using a Sysmex XN-3100 from Japan. PBMC suspensions can be divided into about 140 parts according to 50 µL containing 1.0 × 10^6^ cells. These aliquots were stored in a microcentrifuge tube (Eppendorf Company, Hamburg, Germany) at −80 °C.

### 2.3. Extraction of TAC and MPA from PBMCs in Clinical Samples

The sample preparation was based on paramagnetic beads that remove interfering proteins, phospholipids, and salts from WB, plasma, and serum prior to analysis, precipitate proteins with magnetic separation, and contain the analytes of interest in the supernatant. The sample preparation method was the same for standards, QCs, and patient samples.

First, 30 µL lysis buffer for WB was thoroughly mixed with PBMC samples and left at room temperature for 1 min to completely dissolve the blood cells. Then, 20 µL isotope internal standard (10 ng/mL [^13^C, ^2^H_4_]-tacrolimus and 20 ng/mL MPA-d3), 40 µL MagSiMUS-TDM^prep^ Type I Particle Mix, and 145 µL OPR VI were added to the sample tube at the same time and thoroughly mixed and placed at room temperature for 2 min. The supernatant containing the target analyte was obtained using centrifugation at 15,000× *g* for 5 min (5810R, Eppendorf Company, Hamburg, Germany). Finally, 200 µL of the supernatant was transferred into an automatic sample bottle, and 2 µL was injected into an LC-MS/MS system for liquid phase separation and mass spectrometry analysis. The concentrations of TAC and MPA (ng/mL) in the sample were obtained according to the fitted standard curve, and the obtained concentrations were multiplied by 50 µL to represent the pg content per million cells (pg·10^−6^ cells). The detailed steps for the isolation of PBMCs and the extraction of TAC and MPA from PBMCs in clinical samples are shown in Figure 1.

### 2.4. Preparation of Standards and QC Samples

TAC and MPA initial stock solution were prepared by dissolving accurately weighed drugs in methanol to make a 1 mg/mL solution, and 500 ng/mL working solutions were prepared by diluting the initial stock solution with methanol. The concentration of the isotopic internal standard ([^13^C, ^2^H_4_]-tacrolimus and MPA-d3) stock solution prepared with methanol was 1 mg/mL, and the internal standard stock solution was diluted with methanol to obtain working solutions of 1000 ng/mL and 2000 ng/mL, respectively. Both the stock and working solutions were stored at −80 °C and could be stable for at least 6 months [18]. Standards containing TAC and MPA were prepared using blank PBMC samples and methanol at concentrations ranging from 0.10 ng/mL to 25.00 ng/mL and 0.20 ng/mL to 50.00 ng/mL, respectively. QCs were prepared, stored, and used in the same manner at concentrations of 0.50, 10.00, and 20.00 ng/mL and 1.00, 20.00, and 40.00 ng/mL, respectively. These standards and QCs were stored at −80 °C.

### 2.5. LC-MS/MS System

TAC and MPA were quantified using a Waters Acquity LC-MS/MS system (Waters Corporation, Milford, MA, USA) consisting of liquid chromatography connected to a triple quadrupole Waters TQ-S mass spectrometer, and data processing was performed using the software MasslynxTM V4.1. Chromatographic separation was performed with gradient elution using a reversed-phase Waters Acquity UPLC BEH C18 column (1.7 µm, 50 mm × 2.1 mm). The chromatographic running time was 4.50 min at a column temperature of 50 °C and a flow rate of 0.30 mL/min. Mobile phases A and B were water and methanol, both containing 2 mM ammonium acetate and 0.1% formic acid, as listed in Table 1. MS/MS was run in electrospray ionization (ESI) positive mode with a capillary voltage of 2.50 kV, a desolvation temperature of 550˚C, and argon for collisions. Data acquisition was performed using multiple reaction monitoring (MRM). The product ions (*m*/*z*), cone voltages, and collision energies of TAC, [^13^C, ^2^H_4_]-tacrolimus, MPA, and MPA-d3 are listed in Table 2.

## 3. LC-MS/MS Assay Validation

The assay was fully validated according to the acceptance criteria published by the US Food and Drug Administration (FDA) [22].

### 3.1. Calibration Curve

The calibration curve consisted of a blank sample (without standard and internal standard), a zero sample (blank with internal standard), and 8 non-zero samples covering the expected range. The TAC calibration curve range was 0.10, 0.25, 0.50, 1.00, 2.50, 5.00, 10.00, and 25.00 ng/mL, and the MPA was 0.20, 0.50, 1.00, 2.00, 5.00, 10.00, 20.00, and 50.00 ng/mL. The standards were injected sequentially from low to high concentration, and the concentration was obtained using a weighted (1/x) linear regression of the peak area ratio of TAC/IS, MPA/IS, and concentration. A regression curve was used to calculate intracellular immunosuppressant levels. The curve was considered linear if the correlation coefficient r^2^ was greater than 0.95 and the deviation of the standard from its nominal concentration was less than 15%. A calibration curve, double blank samples, a zero sample, and QC samples were required for each batch of testing.

### 3.2. Selectivity and Specificity

Selectivity and specificity were performed using six blank samples matched to the matrix. The chromatographic peaks of blank and zero samples were required to be undisturbed during the retention time of the standard and internal standard. An acceptable standard was that the internal standard response of the blank sample did not exceed 5% of the average internal standard response of the standards and QCs.

### 3.3. Lower Limit of Quantification

The lowest level of the calibration curve was used as the lower limit of quantification (LLOQ), and performance was assessed by performing six consecutive measurements. Acceptable criteria were an LLOQ signal-to-noise ratio of more than 10, accuracy of 80~120%, and imprecision of less than 20%.

### 3.4. Precision and Accuracy

In order to confirm the precision and accuracy of the assay, low, medium, and high QC levels (QCL, QCM, and QCH) were tested. Intra-batch precision was determined using 6 consecutive tests of QCL, QCM, and QCH in the same batch. Inter-batch precision was completed within 3 days, and QCL, QCM, and QCH were required to be measured simultaneously every day. Acceptable criteria were a coefficient of variation (CV) less than 15% for precision and 85–115% for accuracy.

### 3.5. Matrix and Carryover Effects

Three groups of samples (groups A, B, and C) were prepared, including blank, QCL, and QCH. Group A was prepared by adding the standard into methanol, Group B was prepared by adding the standard into zero samples, and Group C was prepared using a method consistent with that of non-zero samples. Matrix effects were investigated for at least 6 native matrix samples by comparing the ratio of the response of the added standard after matrix extraction to the response of the standard solution. The imprecision in the respective peak areas of the standard and internal standard needed to be less than 15%; otherwise, the method needed to be readjusted to reduce the impact of matrix effects on the imprecision. The matrix effect (B/A, %), recovery (C/B, %) and the extraction effect (C/A, %) should be between 80 and 120%. The evaluation method for the carryover effect included measuring QCL 10 times first, then measuring QCH 10 times, and then measuring QCL 10 times, and then observing whether the relative deviation between the mean of the second batch of QCL and the mean of the first batch of QCL was less than 20%.

### 3.6. Stability

The stability of TAC and MPA was evaluated under different time and temperature conditions. Prior to sample processing, QCL, QCM, and QCH were stored at room temperature for 10 h to assess short-term stability and at −80 °C for 6 months to assess long-term stability. The freeze–thaw stability was evaluated by freezing at −80 °C and thawing at room temperature for 3 times. After sample processing, QCL, QCM, and QCH were stored in the autosampler (8 °C) for 48 h to assess the stability of the samples after extraction. The stability was assessed by calculating QCL, QCM, and QCH concentrations using the newly prepared calibration curve and comparing the mean value of the measured QCs with its nominal concentration with a relative deviation of less than 20%.

### 3.7. Clinical Application

Clinical samples were obtained from 47 patients who received a combination therapy of TAC and MPA after kidney transplantation. Written informed consent was obtained for each patient in accordance with the ethics committee’s instructions. After blood samples were collected, PBMCs were isolated and stored at −80 °C until analysis. The cell density of isolated PBMCs ranged from 0.1 × 10^6^ to 1.0 × 10^6^ cells. To avoid differences between sample cell densities, we normalized the concentrations (ng/mL) obtained from the calibration curve and then conducted statistical analysis. Our normalization method was pg per million cells (pg·10^−6^ cells). PBMC-C_0_ was compared with WB-C_0_, and correlation analyses were performed. WB-C_0TAC_ was obtained using Roche electrochemiluminescence, and C_0MPA_ was obtained using the EMIT method.

### 3.8. Follow-Up Endpoint

Renal transplant recipients were followed up to 6 months after surgery, and the concentrations were monitored dynamically at 5, 7, 14, and 21 days (D5, D7, D14, D21) and 1, 2, 3, and 6 months (M1, M2, M3, and M6), respectively. A composite endpoint was used for follow-up, defined as biopsy-confirmed rejection of kidney transplantation and opportunistic infections (bacterial and viral infections requiring anti-infective therapy) at any time during the 6 months after surgery. Biopsy-confirmed rejection occurred 8 times. Bacterial infections occurred 25 times, including urinary tract infections (56.0%), lung infections (32.0%), bloodstream infections (4.0%), and other infections (8.0%). Virus infections occurred 8 times, including cytomegalovirus infections (62.5%) and BK virus infections (37.5%). Based on whether the above adverse events occurred, kidney transplant recipients were divided into two groups: a stable group (24 cases) and an adverse outcome group (23 cases).

### 3.9. Statistical Analyses

Performance verification data (mean concentration, standard deviation, CV, inaccuracy) were analyzed and calculated using an Excel (2010, Microsoft, Redmond, WA, USA) spreadsheet. GraphPad Prism 8.0 was used to analyze the clinical validation data, that is, to conduct the comparison and correlation analyses of the two groups of data. The method established in this study had detailed experimental procedures, which were the creation of visualization using BioRender.com (https://www.biorender.com, accessed on 15 February 2023) for better and more intuitive understanding. We obtained the publication and licensing of figures for the online site.

## 4. Results

### 4.1. Selectivity and Specificity

In the concentration range of TAC (0.10–25.00 ng/mL) and MPA (0.20–50.00 ng/mL), the calibration curves showed a good linear relationship with the correlation coefficient r^2^ = 0.9988, and r^2^ = 0.9990, respectively. Typical chromatograms of TAC, [^13^C, ^2^H_4_]-tacrolimus, MPA, MPA-d3, and blank extracts are shown in Figure 2. The retention times of TAC and MPA were 3.16 ± 0.08 and 1.80 ± 0.08 min, respectively, and no interfering peaks were observed (Figure 2).

### 4.2. Precision, Accuracy, and LLOQ

Table 3 summarizes the precision and accuracy of LLOQ, QCL, QCM, and QCH for TAC and MPA. The imprecision of QCL, QCM, and QCH was always less than 15% (0.44–3.00%), the accuracy was 85–115% (84.60–110.98%), and the LLOQ was less than 20% (2.13–7.06%).

### 4.3. Matrix and Carryover Effects

The CVs of QCL and QCH for TAC and MPA were less than 15% (0.73–10.05%), and the recovery and extraction effects were also 80–120% (80.03–111.90%), as listed in Table 4. The carryover effect was excellent and up to standard. The matrix effect of MPA met the requirement but was not observed in TAC.

### 4.4. Stability

The stability of TAC and MPA showed that no degradation occurred when QCs were stored at room temperature for 10 h before sample processing or in the automatic sampler for 48 h after sample processing, indicating excellent short-term stability performance. The long-term stability of QCs for MPA was good because there was no obvious degradation after storage at −80 °C for 6 months. However, the long-term stability of TAC was significantly degraded with a relative deviation of more than 20% (28.50–36.00%). TAC degradation was also observed after the third freeze–thaw cycle, while MPA degradation was not observed, indicating that the sample could not be frozen and thawed more than three times during storage and analysis. The stability evaluation results are listed in Table 5.

### 4.5. Clinical Application

This method was used for a total of 47 clinical samples from kidney transplantation patients under combined TAC and MPA treatment for verification. The TAC and MPA concentrations in all patients were within the calibration curve range. These patients had PBMC-C_0TAC_ ranging from 0.12 to 16.40 ng/mL (or 16.05 to 819.90 pg·10^−6^ cells, normalized) and PBMC-C_0MPA_ ranging from 0.51 to 1.01 ng/mL (or 25.40 to 64.21 pg·10^−6^ cells, normalized). The corresponding WB-C_0TAC_ and C_0MPA_ ranged from 3.70 to 13.40 ng/mL and 1.65 to 6.21 ug/mL, respectively.

### 4.6. Follow-Up Outcome

WB-C_0TAC_ decreased, slightly increased (D5-M1), and finally stabilized (M2-M6) (Figure 3A). PBMC-C_0TAC_ first decreased (D5-D14) and then fluctuated around that level (D21-M6) (Figure 3A). After comparing the follow-up time points after renal transplant between stable and adverse outcome groups, no statistical difference was found in WB-C_0TAC_ or PBMC-C_0TAC_, but creatinine (CREA) in the adverse outcome group was significantly higher than that in the stable group in D5-M1 and M6 (Figure 3B–D). The variation difference in WB-IPV and PBMC-IPV of TAC was about 2 times, and the mean value and interquartile interval were 34.61% (21.90, 49.85%) and 77.47% (55.06, 97.76%), respectively (Figure 4A, B). No correlation was found between PBMC-C_0MPA_ and C_0MPA_ from postoperative to M6 (Figure 4C). Preliminary data at the 6-month follow-up showed weak (Figure 5A–E, H) or no correlations (Figure 5F, G) between PBMC-C_0TAC_ and WB-C_0TAC_.

## 5. Discussion

TAC combined with MPA is effective in preventing acute rejection, but its clinical application is complicated by a narrow therapeutic index and significant intra- and inter-patient variability in PK. Therefore, TDM is the most important means for the application of immunosuppressants. Due to the clinical use of WB or plasma concentrations for TDM, renal transplant recipients may still experience rejection and toxicity when monitoring concentrations are maintained within the normal range, which may compromise graft survival [6,23,24]. For example, 85.3% of the TAC in the blood component penetrated into erythrocytes, 14.3% was highly bound to plasma proteins, and 0.46% reached the lymphocytes to exert pharmacological effects, which makes the real exposed TAC concentrations in WB matrix unreliable [9]. Therefore, it is particularly important to develop a method that can simultaneously quantify TAC and MPA in target cells. PBMCs are readily available matrices representing lymphocyte- and monocyte-rich cell populations that have been used in intracellular immunosuppressant concentration assays [12,14,15,16]. Thus, this study established and validated the simultaneous quantification of TAC and MPA concentrations in PBMCs using magnetic bead extraction combined with LC-MS/MS and investigated its value in kidney transplant patients.

Capron, A. et al. first developed a method for quantifying TAC in PBMCs using LC-MS/MS, but the PBMCs isolation and TAC extraction procedures were too laborious and time-consuming to be suitable for routine clinical use [18]. Although other LC-MS/MS methods have been developed in recent years to quantify the content of immunosuppressants in PBMC, these methods still have application limitations and rarely quantify multiple immunosuppressants simultaneously. One method that attracted attention was the use of magnetic beads to adsorb and precipitate proteins and remove impurities using ultracentrifugation to separate TAC, but its disadvantage was that only the concentration of TAC in PBMC was quantified [25]. On the other hand, there were some advantages that were also worth noting. For example, the isolation of PBMC was performed at room temperature without affecting the concentration of TAC, and purification of PBMC was performed by lysing and washing erythrocytes to avoid contamination by TAC bound to erythrocytes. The sample preparation time with MagnaBeads was reduced to less than 10 min. In addition, the chromatographic mass spectrometry operation required a lower injection volume and a very short total running time, and the uniqueness of this method was the use of an isotopic internal standard with the same physical and chemical properties as the internal standard to quantify TAC in PBMC. On the basis of this method, our laboratory established and verified the MagnaBeads combined with LC-MS/MS method for the simultaneous quantitative determination of TAC and MPA concentrations in PBMC.

The performance of the method was evaluated according to FDA guidance [22], and satisfactory results were obtained in terms of specificity, sensitivity, precision, accuracy, and matrix effects. Sensitivity (0.10 ng/mL for TAC and 0.20 ng/mL for MPA, respectively) was high enough to allow reliable quantification of all real samples. Intra-batch and inter-batch imprecision and inaccuracy were less than 15%. It is worth noting that the freeze–thaw stability indicated that the sample could be frozen and thawed no more than three times. The long-term stability of TAC was also degraded, suggesting that PBMC samples could not be frozen at −80 °C for more than 6 months after separation. LC-MS/MS will be a powerful tool for the quantification of intracellular immunosuppressive drugs.

TAC and MPA are effective drugs for the long-term treatment of renal allograft rejection, and clinicians need reliable drug monitoring tools to correlate efficacy and toxicity with drug exposure. Since immunosuppressive drug effects are mediated through inhibition of the lymphocyte proliferation pathway, direct drug quantification in the target region is expected to provide information about drug efficacy more consistent with clinical outcomes [26,27]. Our rapid and sensitive method was applied to PBMC samples from 47 kidney transplantation patients. Our results showed that IPV reflected large PK differences, and the value of PBMC-IPV was higher and should receive more attention in the process of individual precision medicine. Although there was no difference in PBMC-C_0TAC_ between stable and adverse outcome groups, the dynamic change process showed that the fluctuation in PBMC-C_0TAC_ was significantly higher than in WB-C_0TAC_, which was because the level of WB-C_0TAC_ was paid more attention in clinical practice. For TAC, weak correlations were observed between intracellular and conventional blood concentrations, suggesting that quantitative monitoring of intracellular immunosuppressive concentrations was necessary. The results were also found in other experiments [11,12,14,27].

A 12-month prospective PK study showed that the PK exposure parameters of TAC and MPA were time-dependently associated with specific drug-induced toxicity or acute rejection [28]. PK studies in the first year after renal transplantation showed that the distribution dynamics of intracellular TAC were altered [29]. The mechanism or clinical significance of the kinetic change remains unknown and requires further investigation. ABCB1, present in the monocyte membrane, is an efflux transporter, and TAC is a substrate of ABCB1. Individual differences in ABCB1 activity may lead to differences in TAC distribution and may affect drug efficacy [30,31].

The PK of MPA is characterized by high inter- and intra-individual variability, and its exposure is significantly influenced by renal function, albumin levels, hemoglobin, and dose of CsA [32]. MPA is a selective inhibitor of inosine monophosphate dehydrogenase (IMPDH), an enzyme involved in the intracellular synthesis pathway of T and B lymphocytes. During kidney transplantation, PBMC-C_0MPA_ did not correlate with IMPDH activity, which may be determined using inter-patient variability [33]. Longitudinal measurements of IMPDH and purine levels in PBMC in the first year after kidney transplantation suggested that the molecular PD response of MPA was more inhibited on activated lymphocytes than on resting lymphocytes, with potential applicability in patients at risk of MPA overexposure [34]. Based on the rationale that monitoring target enzyme activity may reflect the efficacy of immunosuppression, PD monitoring of IMPDH has been suggested as a complementary approach to individualized MPA therapy.

## 6. Conclusions

The proposed method was fast, accurate, and precise, and can be used for conventional TDM of intracellular TAC and MPA in the future, providing clinicians with more reliable exposure information of TAC and MPA in target cells. We designed a prospective follow-up study to more comprehensively and accurately evaluate the correlation between concentrations in PBMCs and WBs, which can make up for the small number of clinically validated patients in this study. However, the relationship between PBMC-C_0TAC_ and adverse outcomes of renal transplant needs further study, and the PK and PD of intracellular immunosuppressive agents still need more investigation.

## Figures and Tables

**Figure 1 pharmaceutics-15-02318-f001:**
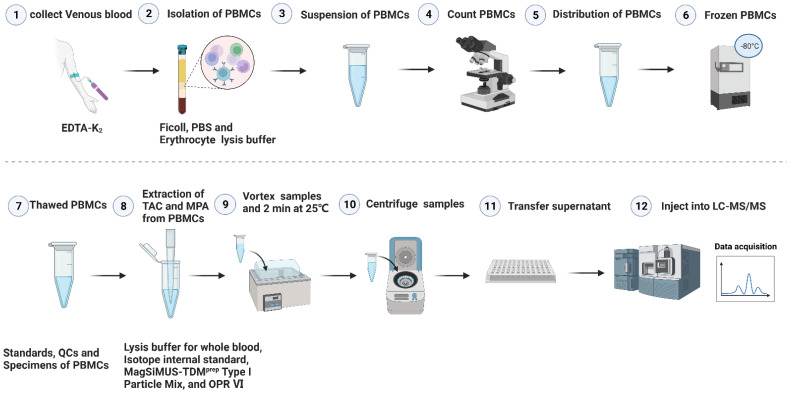
The work content of this study presented in visual form. The detailed content is divided into 12 sections as shown in the figure. In this study, we developed and established a reliable method for monitoring TAC and MPA concentrations in PBMC using single sample analysis, which provided a research basis for exploring intracellular immunosuppressive concentration, and then provided methodological support for postoperative TDM management and long-term follow-up efficacy evaluation of renal transplant recipients.

**Figure 2 pharmaceutics-15-02318-f002:**
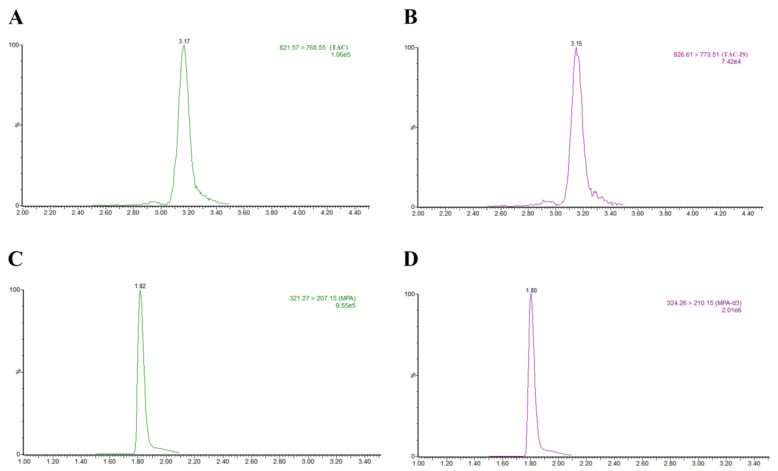
Typical ion chromatograms of PBMC extracts obtained using simultaneously spiked samples with TAC and ^13^C, ^2^H_4_- tacrolimus, MPA, and MPA-d3. (**A**,**B**) are typical ion chromatograms of TAC and ^13^C, ^2^H_4_- tacrolimus, respectively. (**C**,**D**) are typical ion chromatograms of MPA and MPA-d3, respectively. Note: The response values of typical ion chromatographic peaks were: 1.06 × 10^5^, 7.42 × 10^4^, 9.55 × 10^5^ and 2.01 × 10^6^.

**Figure 3 pharmaceutics-15-02318-f003:**
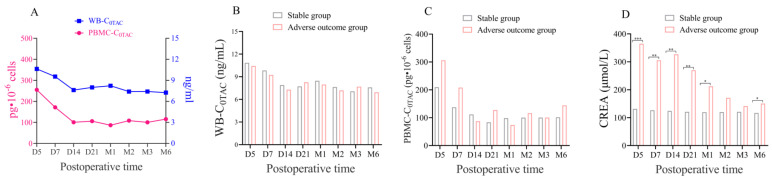
(**A**) The dynamic changes in WB-C_0TAC_ and PBMC-C_0TAC_ from D5 to M6 after renal transplant. (**B**–**D**) A comparison of WB-C_0TAC_, PBMC-C_0TAC_, and CREA between the stable group and adverse outcome group, respectively. Note: These symbols indicated statistical differences. * was *p* < 0.05, ** was *p* < 0.01, *** was *p* < 0.001.

**Figure 4 pharmaceutics-15-02318-f004:**
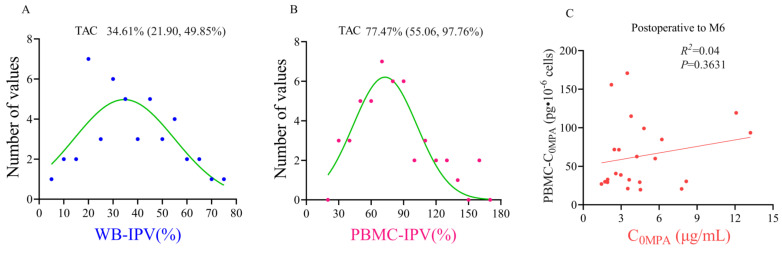
(**A**) WB-IPV frequency distribution histogram of TAC. (**B**) PBMC-IPV frequency distribution histogram of TAC. The mean value and interquartile interval were used to describe the results of (**A**,**B**). (**C**) Correlation analysis between PBMC-C_0MPA_ and C_0MPA_ from postoperative to M6.

**Figure 5 pharmaceutics-15-02318-f005:**
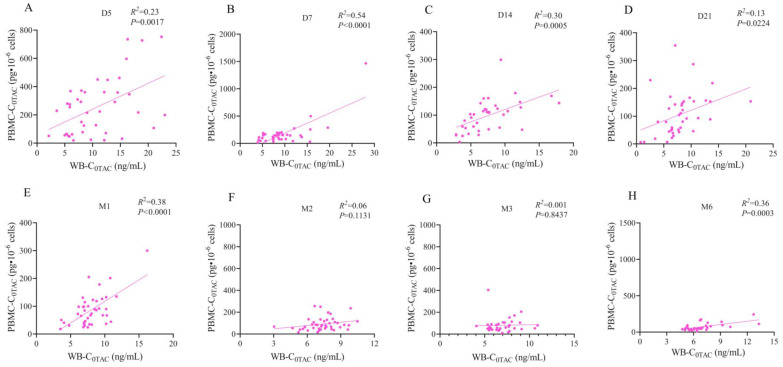
Correlation analysis between PBMC-C_0TAC_ and WB-C_0TAC_ in 47 renal transplant patients. The follow-up time was D5, D7, D14, D21, M1, M2, M3, and M6, respectively. (**A**) is significantly correlated at D5 (*R*^2^ = 0.23, *p* = 0.0017). (**B**) is significantly correlated at D7 (*R*^2^ = 0.54, *p* < 0.0001). (**C**) is significantly correlated at D14 (*R*^2^ = 0.30, *p* = 0.0005). (**D**) is significantly correlated at D21 (*R*^2^ = 0.13, *p* = 0.0224). (**E**) is significantly correlated at M1 (*R*^2^ = 0.38, *p* < 0.0001). (**F**) is irrelevant at M2 (*R*^2^ = 0.06, *p* = 0.1131). (**G**) is irrelevant at M3 (*R*^2^ = 0.001, *p* = 0.8437). (**H**) is significantly correlated at M6 (*R*^2^ = 0.36, *p* = 0.0003).

**Table 1 pharmaceutics-15-02318-t001:** Chromatographic gradient: the mobile phases A and B were water and methanol, respectively, both containing 2 mM ammonium acetate and 0.1% formic acid.

Time (min)	Flow (mL/min)	Mobile Phase A (%)	Mobile Phase B (%)
Initial	0.30	50.00	50.00
0.50	0.30	50.00	50.00
3.00	0.30	5.00	95.00
4.50	0.30	50.00	50.00

**Table 2 pharmaceutics-15-02318-t002:** Optimized settings for multiple reaction monitoring (MRM) for each analyte.

Analyte	ESI Mode	Parent Ion (*m*/*z*)	Product Ion (*m*/*z*)	Cone Voltage (V)	Collision Energy (eV)
TAC	+	821.57	768.55	24.00	18.00
[^13^C, ^2^H_4_]-tacrolimus	+	826.61	773.51	14.00	18.00
MPA	+	321.27	207.15	10.00	22.00
MPA-d3	+	324.26	210.15	8.00	22.00

**Table 3 pharmaceutics-15-02318-t003:** Intra/inter-batch precision and accuracy of TAC and MPA at LLOQ, QCL, QCM, and QCH concentrations.

Concentration (ng/mL)		Intra-Batch			Inter-Batch		
		Found (ng/mL)	Imprecision CV (%)	Accuracy (%)	Found (ng/mL)	Imprecision CV (%)	Accuracy (%)
TAC *	0.10	0.09	2.13	90.00			
	0.50	0.46	2.19	92.00	0.45	3.00	90.00
	10.00	8.55	0.44	85.50	8.46	0.95	84.60
	20.00	17.78	0.84	88.90	17.71	1.23	88.55
MPA *	0.20	0.20	7.06	100.00			
	1.00	0.96	1.63	96.00	0.94	1.84	94.00
	20.00	19.09	0.53	94.45	18.96	1.35	94.80
	40.00	44.39	0.78	110.98	43.60	1.05	109.00

* For TAC, the concentrations of LLOQ, QCL, QCM, and QCH were 0.10, 0.50, 10.00, and 20.00 ng/mL, respectively. For MPA, the concentrations of LLOQ, QCL, QCM, and QCH were 0.20, 1.00, 20.00, and 40.00 ng/mL, respectively.

**Table 4 pharmaceutics-15-02318-t004:** Matrix, recovery, and extraction effects of TAC and MPA.

Concentration (ng/mL)		Groups	Found (ng/mL)	CV (%)	Accuracy (%)	Matrix Effect (B/A, %)	Recovery (C/B, %)	Extraction Effect (C/A, %)
TAC	0.50	A	0.42	0.73	84.00	130.95	85.45	111.90
		B	0.55	4.21	110.00			
		C	0.47	10.05	94.00			
	20.00	A	16.49	2.14	82.45	124.20	80.03	99.39
		B	20.48	1.77	102.40			
		C	16.39	3.71	81.95			
MPA	1.00	A	1.20	1.68	120.00	105.00	92.86	97.50
		B	1.26	3.18	126.00			
		C	1.17	5.44	117.00			
	40.00	A	38.87	8.83	97.18	116.72	83.93	97.97
		B	45.37	1.44	113.43			
		C	38.08	3.28	95.20			

**Table 5 pharmaceutics-15-02318-t005:** Stability evaluation of TAC and MPA.

Concentration (ng/mL)		10 h before Sample Processing		48 h after Sample Processing		Long-Term Stability		Freeze–Thaw Stability	
		CV (%)	Accuracy (%)	CV (%)	Accuracy (%)	CV (%)	Accuracy (%)	CV (%)	Accuracy (%)
TAC	0.50	9.25	96.00	4.69	90.00	2.22	64.00	4.59	66.00
	10.00	0.64	84.40	0.70	84.60	0.66	71.50	1.64	81.50
	20.00	0.73	87.55	0.73	87.90	0.64	68.75	0.65	81.30
MPA	1.00	1.07	95.00	1.95	94.00	1.59	90.00	2.72	98.00
	20.00	0.56	93.15	0.83	93.70	0.50	82.65	0.92	101.35
	40.00	0.54	108.45	0.73	108.28	0.76	104.53	0.35	99.25

## Data Availability

The data are presented in this article. For more information, please contact corresponding the author, Y.L. (Yi Li).

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
