# Peer review of "Dynamic Monitoring of Intracellular Tacrolimus and Mycophenolic Acid Therapy in Renal Transplant Recipients Using Magnetic Bead Extraction Combined with LC-MS/MS"

_pharmaceutics, 2023, doi:10.3390/pharmaceutics15092318_

Round 1

Reviewer 1 Report

The manuscript by Xu et al. entitled "Dynamic Monitoring of Intracellular Tacrolimus and Mycophenolic Acid Therapy in Renal Transplant Recipients by Magnetic Bead Extraction Combined with LC-MS/MS" describes the development of a clinical method for monitoring drug levels in PBMCs of renal transplant patients by LC-MS/MS, with samples prepared using a magnetic bead-based protocol. The paper is quite polished and its results would be valuable for the field. I only see a couple small points to address prior to publication.

Page 4, line 153 - it would be more useful for study replication attempts to provide the centrifugation speed in g rather than rpm.

Page 6, line 249 - was the autosampler refrigerated or kept at room temperature?

Page 9, line 339 - please provide expansion for the CREA acronym where it first appears in the text.

Page 10, figure 5 - there are several clear outliers present (particularly clear in F, G and H). In panel H especially, the outlier appears to skew the trendline upwards. Do these outliers originate from related samples (e.g. same patient)? Do the authors have an explanation for their presence? If these are removed, are the conclusions affected?

Author Response

Response to Reviewer 1 Comments

1. Summary

Thank you very much for taking the time to review this manuscript. Your valuable comments were very helpful to us, especially about whether to reanalyze the data after removing the outliers. After our verification, it was agreed that it was more reasonable to perform data analysis after removing outliers. Please find below the detailed responses and corresponding revisions/corrections made in the manuscript.

2. Questions for General Evaluation

Reviewer’s Evaluation

Response and Revisions

Does the introduction provide sufficient background and include all relevant references?

Yes

Are all the cited references relevant to the research?

Yes

Is the research design appropriate?

Yes

Are the methods adequately described?

Yes

Are the results clearly presented?

Yes

Are the conclusions supported by the results?

Yes

3. Point-by-point response to Comments and Suggestions for Authors

Comments 1: Page 4, line 153 - it would be more useful for study replication attempts to provide the centrifugation speed in g rather than rpm.

Response 1: Thank you for pointing this out. We agree with this comment. Therefore, we have used g as the unit of centrifugation speed to ensure consistency throughout the text.

“[The supernatant containing the target analyte was obtained by centrifugation at 15,000 g for 5 min (5810R, Eppendorf company, Germany).]”- Page 4, line 152 and 153

Comments 2: Page 6, line 249 - was the autosampler refrigerated or kept at room temperature?

Response 2: Thank you for pointing this out. The temperature of the autosampler was set at 8°C, which was refrigerated.

“[After sample processing, QCL, QCM and QCH were stored in the autosampler (8℃) for 48 hours to assess the stability of the samples after extraction.]” - Page 6, line 249

Comments 3: Page 9, line 339 - please provide expansion for the CREA acronym where it first appears in the text.

Response 3: Thank you for pointing this out. This was an oversight on our part, and CREA is a generic acronym for creatinine. We have made additions and revisions in the text.

“[but the creatinine (CREA) in the adverse outcome group was significantly higher than that in the stable group in D5-M1 and M6 (Figure 3B~D).]” - Page 9, line 339

Comments 4: Page 10, figure 5 - there are several clear outliers present (particularly clear in F, G and H). In panel H especially, the outlier appears to skew the trendline upwards. Do these outliers originate from related samples (e.g. same patient)? Do the authors have an explanation for their presence? If these are removed, are the conclusions affected?

Response 4: Thank you for pointing this out. After checking the original data, the outlier appearing in Figure 5F was indeed high. To reduce confusion for readers, we decided to rerun the analysis after removing outlier, and the result showed that there was still no correlation. The abnormal values in Figure 5G and H were measured at M3 and M6 respectively in the same patient, which may exist during the sample retention process, because the source of the transplant recipient’s kidney was a deceased patient, and the dose of TAC can be adjusted according to the condition. In order to avoid outliers affecting the overall analysis results, we decided to adopt the reviewer’s suggestion and reanalyze the data after removing the outliers. The results showed that there was no correlation in Figure 5G, and the correlation in Figure 5H was weak. The results after removal of outliers did not affect the conclusions of the study.

Preliminary data at 6-month follow-up showed weak (Figure 5A~E and H) or no correlations (Figure 5F and G) between PBMC-C0TAC and WB-C0TAC.]” - Page 9, line 344 and 345

“[F was irrelevant at M2 (R2=0.06, P=0.1131). G was irrelevant at M3 (R2=0.001, P=0.8437). H was significantly correlated at M6 (R2=0.36, P=0.0003).]” - Page 11, line 361 and 362

4. Response to Comments on the Quality of English Language

Point 1: English language fine. No issues detected

Response 1: Thank you for the reviewer's assessment of my English language quality. I will urge myself to take a step further in the future academic research.

5. Additional clarifications

Thanks to the journal editor/reviewer for editing and commenting on our manuscript, we only have three clarification at this time.

Page 3, line 133- The centrifuge model used in the process of separating PBMCs is 5810, Eppendorf company, Germany.

Page 3, line 135 and 136-We added a space in the middle of 300g, which has been marked in red in the manuscript.

Page 12, line 454, 460, 461 and 462- Supplement a summary statement in the "Author Contributions" section of the manuscript text that reflected the contributions of all authors involved.

Reviewer 2 Report

 The presented study is relevant in terms of monitoring the effectiveness of immunosuppressants used in kidney transplantation. Can be published as presented, but there are a few clarification questions.

1.It should be noted that the patient's peripheral blood sample was taken at what time after transplantation and administration of immunosuppressants

2.It is not entirely clear how the content of immunosuppressors in the blood cells of voluntary healthy donors was compared?

 3.Since immunosuppressants were administered to patients, did you check the stability of TAC and MPA at 370 C?

Author Response

Response to Reviewer 2 Comments

1. Summary

Thank you very much for taking the time to review this manuscript. Your valuable comments were very helpful to us, please find the corresponding responses below for questions that need clarification.

2. Questions for General Evaluation

Reviewer’s Evaluation

Response and Revisions

Does the introduction provide sufficient background and include all relevant references?

Yes

Are all the cited references relevant to the research?

Yes

Is the research design appropriate?

Yes

Are the methods adequately described?

Yes

Are the results clearly presented?

Yes

Are the conclusions supported by the results?

Yes

3. Point-by-point response to Comments and Suggestions for Authors

Comments 1: It should be noted that the patient's peripheral blood sample was taken at what time after transplantation and administration of immunosuppressants.

Response 1: Thank you for pointing this out. Peripheral blood samples from patients were collected on day 5 after transplantation, which was day 3 of immunosuppressant administration. Each blood collection time was collected half an hour before taking the medicine in the morning.

Comments 2: It is not entirely clear how the content of immunosuppressors in the blood cells of voluntary healthy donors was compared?

Response 2: Thank you for pointing this out. Firstly, PBMCs isolated from the peripheral blood of healthy volunteers do not contain immunosuppressants and can be used as blank matrix. Then we mixed the working solution containing the immunosuppressant with the blank matrix in volume ratio, and through the same step of dissolving the cell membrane to release the intracellular immunosuppressant, the same environment as that of the patient's PBMCs could be obtained. Finally, the supernatant after magnetic bead separation and protein precipitation was obtained by the same processing steps, and the immunosuppressant content in the supernatant was measured by LC-MS/MS.

Comments 3: Since immunosuppressants were administered to patients, did you check the stability of TAC and MPA at 37℃?

Response 3: Thank you for pointing this out. In the data previously published by our research team, the stability of TAC and MPA at 37°C was proved. The results showed that "the maximum average deviation of TAC was 13.49% and MPA was 8.90% when placed at 37°C for 7 days", both of which were within the acceptable range.

4. Response to Comments on the Quality of English Language

Point 1: I am not qualified to assess the quality of English in this paper.

Response 1: Thank you for the reviewer's feedback on my English language quality. This shows that there is still a lot of room for improvement in my English language, and I will continue to learn English expressions.

5. Additional clarifications

Thanks to the journal editor/reviewer for editing and commenting on our manuscript, we have no additional clarifications at this time.

Reviewer 3 Report

The article is within the scope of the journal and presents findings of research titled “Dynamic Monitoring of Intracellular Tacrolimus and Mycophenolic Acid Therapy in Renal Transplant Recipients by Magnetic Bead Extraction Combined with LC-MS/MSon work studied by the standard methodology. The paper presents very interesting results as well as an inquisitive and reliable interpretation of the research results. The topic original and relevant in the field of study. Researchers evaluated and validated the performance of the methodology to quantify TAC and MPA concentrations in peripheral blood mononuclear cell (PBMC) using liquid chromatography tandem mass spectrometry (LC-MS/MS), and concluded that his rapid and sensitive method can provide clinically more reliable information of drug concentration at the active site, which has the potential to be applied to the clinical monitoring of intracellular immunosuppressive concentration in organ transplantation. The methodology adequately described and conclusion consistent with the evidence and arguments presented

Minor typographical and grammatical errors need addressing.

Author Response

Response to Reviewer 3 Comments

1. Summary

Thank you very much for taking the time to review this manuscript. We are pleased to receive the reviewers' affirmation of the study design and results, and we will continue to explore more interesting topics in this research area.

2. Questions for General Evaluation

Reviewer’s Evaluation

Response and Revisions

Does the introduction provide sufficient background and include all relevant references?

Yes

Are all the cited references relevant to the research?

Yes

Is the research design appropriate?

Yes

Are the methods adequately described?

Yes

Are the results clearly presented?

Yes

Are the conclusions supported by the results?

Yes

3. Point-by-point response to Comments and Suggestions for Authors

Comments 1: Comments on the Quality of English Language

Minor typographical and grammatical errors need addressing.

Response 1: Thank you for pointing this out. We reorganized the English language expression of the full text of the manuscript and found no special changes. Thanks again to the reviewer for valuable comments, and we will pay more attention to the expression of English language in future academic research.

4. Response to Comments on the Quality of English Language

Point 1: Minor editing of English language required.

Response 1: Thank you for the reviewer's assessment of my English language quality. This shows that there is still a lot of room for improvement in my English language, and I will continue to learn English expressions.

5. Additional clarifications

Thanks to the journal editor/reviewer for editing and commenting on our manuscript, we have no additional clarifications at this time.
